# Niche partitioning facilitates coexistence of three apex predators in the Lapchi Valley, Central Himalaya, Nepal

Narayan Prasad Koju[1,2,3]*, Babita Maharjan[4,5], Meghajan Budha[6], Ashim Thapamagar[1], Nishan Pokhrel[6], Miriam Lee[7,8], Laxman Khanal[4], Paul Buzzard[9], Xuelong Jiang[2], Randall C. Kyes[3,10]

**1** Center for Postgraduate Studies, Nepal Engineering College, Pokhara University, Lalitpur, Nepal, **2** Key Laboratory of Genetic Evolution and Animal Models, Kunming Institute of Zoology, Chinese Academy of Sciences, Kunming, Yunnan, China, **3** Department of Psychology, University of Washington, Seattle, Washington, United States of America, **4** Central Department of Zoology, Institute of Science and Technology, Tribhuvan University, Kathmandu, Nepal, **5** Amrit Campus, Tribhuvan University, Thamel, Kathmandu, Nepal, **6** Goldengate International College, Tribhuvan University, Battisputali, Kathmandu, Nepal, **7** School of Architecture, The Chinese University of Hong Kong, Hong Kong, China, **8** China Exploration and Research Society, Hong Kong, **9** Washtenaw County Conservation District, United States of America, **10** Departments of Global Health and Anthropology, Center for Global Field Study, Washington National Biomedical Research Center, University of Washington, Seattle, Washington, United States of America

* npkoju.2003@gmail.com

## Abstract

Large carnivores increasingly coexist in human-modified mountain landscapes, yet empirical evidence on how multiple apex predators partition space, time, and resources to reduce competition remains limited, particularly in the Central Himalaya, Nepal. Understanding these mechanisms is critical for predicting interspecific interactions and informing conservation practices in resource-limited alpine ecosystems. This study explored the ecological interactions, competitive dynamics, and coexistence strategies of sympatric snow leopards (*Panthera uncia*), leopards (*Panthera pardus*), and Himalayan wolves (*Canis lupus chanco*) in the Lapchi Valley, Central Himalaya, Nepal. We examined spatial distribution, temporal activity patterns, and dietary overlap among these apex predators using a combination of camera trapping, scat-based DNA analysis, and micro-histological diet assessment. The results showed a complete (100%) spatial overlap between the snow leopards and wolves' ranges. All three predators exhibited predominantly nocturnal activity with strong temporal overlap (0.78). Dietary analyses showed a clear trophic segregation: snow leopards relied mainly on wild ungulates, leopards consumed synanthropic prey, while wolves consumed a mixed diet combining wild and domestic prey. Pianka's index indicated high dietary overlap between snow leopards and wolves (0.77), but remarkably low overlap of these predators with leopards. The multidimensional niche partitioning appears to reduce direct competition among predators. These findings highlight the role of behavioral flexibility, spatial segregation, and prey selection

**Data availability statement:** The supporting information includes: supplementary figure, supplementary tables and overlapR code are available at https://doi.org/10.5281/zenodo.18053588.

**Funding:** This study was supported by a faculty grant from the University Grants Commission (UGC), Nepal (FRG 77/78 S&T-01 2020/21 and FRG 080/81 S&T-07), Andrew Sabin Family Foundation, Zoo New England, and the China Exploration and Research Society, Hong Kong. RCK's effort was supported in part by the Office of Research Infrastructure Programs (ORIP) of the National Institutes of Health through grant number P51OD010425 to the Washington National Biomedical Research Center, USA.

**Competing interests:** We declare no conflict of interest.

in promoting the coexistence of predators. Conservation strategies must prioritize sustaining wild prey populations, mitigating livestock depredation, and addressing climate-driven habitat shifts that may intensify interspecific competition.

## 1. Introduction

Apex predators play a critical role in shaping terrestrial ecosystems through regulation of prey populations and cascading trophic effects [1,2]. In the Himalayas, snow leopards (*Panthera uncia*) and Himalayan wolves (*Canis lupus chanco*) are the dominant alpine carnivores, occupying alpine and subalpine habitats across the high-elevation landscapes of the central Himalaya. Snow leopards, a flagship species of high-altitude ecosystems, occur in fragmented populations across Central Asia, with Nepal representing a globally significant stronghold [3,4]. Recent surveys estimate a population of approximately 397 individuals in Nepal, highlighting the species' vulnerability and the country's conservation responsibility [5]. Himalayan wolves occupy alpine and subalpine habitats that overlap with snow leopards, and their flexible foraging strategies allow them to persist in landscapes heavily influenced by pastoralism.

In recent years leopards (*Panthera pardus*) have increasingly expanded into high-elevation habitats traditionally occupied by snow leopards [6–8]. Leopards are ecological generalists inhabiting forests, grasslands, and alpine zones, facilitating their upward expansion into snow leopard range. This recent range expansion has created novel zones of sympatry among three similarly sized apex predators with overlapping diets in particular, thereby increasing the potential for interspecific competition. Snow leopards and leopards are classified as Vulnerable, and Himalayan wolves are Least Concern according to IUCN Red List. However, all three species face ongoing threats from prey depletion, habitat fragmentation, and wildlife conflict [9,10].

Understanding how these sympatric apex predators partition ecological niches and coexist under increasing anthropogenic pressure is therefore critical, particularly in the resource limited alpine habitat of the central Himalaya. Despite increasing spatial overlap and shared exposure to anthropogenic and climatic pressures, empirical evidence on the mechanisms enabling coexistence among these three apex predators in the central Himalaya remains limited [6,11–13].

Interspecific interactions among large carnivores are complex, involving competition for prey and space, behavioral interference, kleptoparasitism, and even lethal encounters [14,15]. Studies from China demonstrate spatial and dietary overlap between leopards and snow leopards, and Nepalese camera trap surveys confirm sympatry in Annapurna Conservation Area, Gaurishankar Conservation Area, and Sagarmatha National Park [6,8,16,17]. Spatial segregation often facilitates coexistence despite high dietary overlap, but recent Himalayan wolf recolonization and climate-mediated habitat changes may intensify competition. Leopard expansion into alpine zones may further displace snow leopards, with cascading effects on prey populations and ecosystem dynamics [6,7].

The Lapchi Valley in Gaurishankar Conservation Area, Nepal has emerged as a critical site for investigating these dynamics. Initial surveys documented only snow leopards [18], but recent evidence confirms leopard increasing spatio-temporal overlaps [6] and Himalayan wolf colonization. This convergence offers a rare opportunity to investigate how spatial use, activity patterns, and dietary strategies interact to facilitate coexistence under conditions of environmental and anthropogenic change.

Guided by ecological theory on niche partitioning and intraguild competition, this study explicitly tests hypotheses regarding the mechanisms enabling coexistence among snow leopards, leopards, and Himalayan wolves in the Lapchi Valley. We hypothesized that coexistence among these apex predators is facilitated primarily through trophic niche differentiation rather than strong spatial or temporal segregation. Specifically, we predicted that (i) snow leopards and Himalayan wolves would exhibit substantial spatial overlap due to shared use of alpine and subalpine habitats, whereas leopards would show partial spatial segregation by concentrating activity at lower elevations and closer to human settlements; (ii) all three predators would display predominantly nocturnal activity, resulting in high temporal overlap and limited time-based segregation; and (iii) dietary differentiation would be pronounced, with snow leopards specializing on wild alpine ungulates, leopards relying more heavily on livestock and synanthropic prey, and Himalayan wolves exhibiting dietary plasticity by exploiting both wild and domestic prey. We further predicted that dietary overlap would be greatest between snow leopards and Himalayan wolves, reflecting their greater ecological similarity and potential for exploitative competition.

By testing these predictions using long-term camera trapping and scat-based dietary analyses, this study aims to clarify the ecological mechanisms that structure large carnivore assemblages in the central Himalaya. It also aimed to inform conservation strategies that balance apex predator persistence with the livelihoods of pastoral communities in rapidly changing mountain landscapes.

## 2. Materials and methods

### 2.1 Study area

The Gaurishankar Conservation Area (GCA), located in Nepal's central Himalayas, spans 2,179 km² and encompasses one of the world's most climatically diverse landscapes, ranging from sub-tropical to nival zones [18,19]. The area supports 16 major vegetation types and rich faunal diversity, including 235 birds, 77 mammals, 16 fish, 22 reptiles, and 10 amphibians [20]. GCA is a key corridor for snow leopards, connecting the Tibetan Plateau to the north with Langtang and Sagarmatha National Parks to the west and east, respectively [13].

Lapchi Valley, part of the Lamabagar-Lapchi block, covers ~180 km² and extends north into Tibet, such that it is bordered on three sides by China (Fig 1). The valley ranges in elevation from 968 to 7,181 masl and experiences summer monsoon rains (June–August) and winter snowfall (January–March) [21]. Revered as a *'Beyul"* by Vajrayana Buddhists, Lapchi is a spiritual refuge where local communities uphold environmental taboos and wildlife protection through ritual practices. Situated at the base of the Lapchi Khang range, the valley is a prominent pilgrimage site centered on ChöraGephel Ling monastery and meditation caves associated with the saint-poet Jetsun Milarepa. This religious significance fosters harmonious coexistence between humans and wildlife, despite occasional livestock predation [22]. The community in Lapchi raises yaks and horses and practices semi-nomadic pastoralism, moving yaks and horses seasonally between high and low pastures.

### 2.2 Data collection

**2.2.1 Camera trapping.** To investigate habitat utilization, daily activity patterns, and spatiotemporal interactions among the leopards, snow leopards, Himalayan wolves and their potential prey (including domestic livestock), a camera trapping (CT) survey was conducted. The survey spanned from October 2018 to March 2025 and occurred in three phases. Phase I was conducted from October 22, 2018 to May 16, 2019. Phase II took place from October 19, 2021

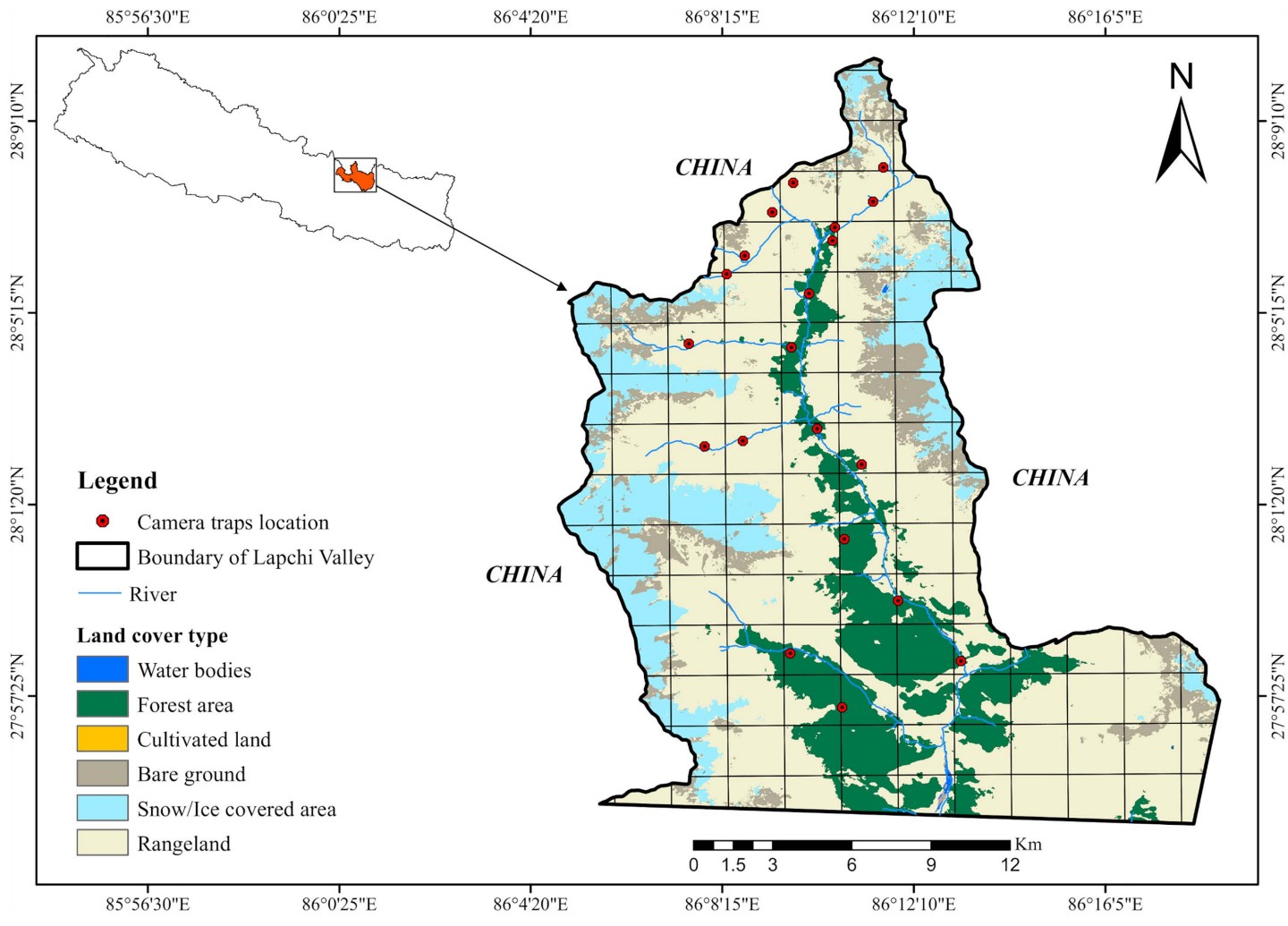

**Fig 1. Lapchi Valley study area, showing locations of camera trap (CT) placement.**

to March 15, 2023. Phase III was conducted from October 11, 2024 to March 14, 2025. The data were divided into two seasons. The cold (winter) season was from November 16 to April 15, while the warm (summer) season spanned from April 16 to November 15 following Koju et al., (6). The total camera trapping effort comprised 9,742 trap days.

A total of 26 CTs were deployed across the study landscape in twenty locations. The survey grid was stratified using 2 × 2 km² cells, and CTs were allocated to cells with an inter-station distance of at least 2 km to reduce spatial autocorrelation between elevational range of 2,200 masl −4,700 masl. Each CT unit was positioned at a height of 30–40 cm above the ground, with the exact placement determined by local topography and slope. Installation protocols followed established guidelines [23]. To minimize false triggers, sensors were oriented away from direct sunlight [24]. We used Bushnell Trophy Cam HD Trail Cameras (Essential E3 16MP and E2 12MP), set in hybrid mode to capture three photos followed by a 10-second video, with sensitivity set to low. Both images and videos were integrated into the analysis. While cameras operated continuously, deployment periods varied across locations and years. The trigger interval between images was standardized at one second, ensuring comprehensive documentation of passing individuals. All CTs operated continuously (24 h/day), thereby enabling monitoring of both diurnal and nocturnal behavior.

**2.2.2 Scat collection.** Fresh scat samples of the three-predator species were collected during seven field study periods: October 2021, April and November 2022, June and November 2024, January and March 2025. In total, 182 putative scats samples were collected. For each sample 10–15 gm of fecal materials was placed in a sterile plastic bag with silica crystals. Additionally, a small fragment of the scats were collected in 2 ml tube containing lysis buffer for subsequent genetic analyses, following protocols outlined by Janečka et al., (25). All samples were collected following established protocols by Panthera and the Snow Leopard Trust [4] and PAWs [25]. A portion of the scat was left in the field in its original location to minimize disruption to the regular movements and territorial marking of the predators [26]. All samples were taken to the Molecular Laboratory of the Central Department of Zoology (CDZ), Tribhuvan University and stored at −20°C.

**2.2.3 DNA extraction.** DNA extraction was performed on the scat samples (collected in lysis buffer) using the QIAamp DNA Stool Mini Kit (Qiagen, Germany), following a modified protocol. Samples were homogenized in ASL lysis buffer, followed by the addition of an InhibitEX tablet to minimize PCR inhibitors. After centrifugation, protein digestion was performed with Proteinase K, and the lysate was treated with Buffer AL before incubation at 70°C for 10 minutes to ensure complete cell lysis. DNA was eluted in 20 µl and stored for further analyses. From the 182 scat samples collected, 93 samples had adequate quantities of DNA for further laboratory analysis.

**2.2.4 Predator identification.** A hierarchical multiplex PCR approach was implemented to determine the predator species identity from the scat samples. In the first stage, all samples were screened using a general carnivore primer pair (CYTB-SCT-F/R) targeting the mitochondrial cytochrome b (Cytb) gene [27]. Each PCR reaction contained standard buffer, $MgCl_2$, dNTPs, BSA, Amplitaq polymerase, forward and reverse primers, distilled water, and DNA template. Amplification was performed over 50 thermal cycles, and products were examined on 2% agarose gels to verify successful amplification.

Samples that produced positive amplification were subsequently subjected to species-specific identification. Snow leopard DNA was detected using Cytb-targeted primers (CYTB-SCT-PUN-F/R) developed by Janečka et al., [28]. PCR amplification was carried out for 50 cycles with an annealing temperature of 60 °C, and products were visualized by gel electrophoresis. A reference DNA sample for a snow leopard and a leopard, previously confirmed through d-loop sequencing, were included as a positive control in all reactions. Samples that tested negative for snow leopard were further PCR amplified using NADH4-targeted primers (NADH4-F/R) for leopard following the protocol of Mondol et al., [29], running 50 cycles with an annealing temperature of 56.5°C.

To further validate species identification, all remaining unidentified samples, along with a subset of three previously confirmed snow leopard and three leopard samples, were analyzed using primers targeting the mitochondrial d-loop region. This additional step served both to confirm ambiguous cases and to independently verify the specificity of the snow leopard and leopard primer sets. Of the 93 DNA extracts analyzed, 14 were confirmed as snow leopard, 11 as leopard, and 18 as Himalayan wolf. In addition, 18 samples were identified as red fox, 13 as leopard cat, while 19 samples failed to amplify with the primers.

**2.2.5 Micro-histological diet analysis.** After predator identity confirmation, a subset of scat samples underwent micro-histological examination to determine prey composition. Metabarcoding and other molecular tools could not be conducted due to facility limitations in Nepal. The protocol for identification followed the guideline of Oli et al., [30] with refinements from Mukherjee et al., [31].

Scat samples were first washed through fine mesh sieves to retain undigested remains (e.g., bones, teeth, hairs). Samples were further treated with carbon tetrachloride ($CCl_4$) and oven-dried at 60°C. From each scat, 20 hairs were randomly selected using a randomized grid method to avoid observer bias. Each hair was mounted in DPX medium under coverslips and examined under a light microscope (400 × magnification). Prey species were identified by comparing medullary structures against reference hair slides prepared from local wild and domestic fauna. Where reference slides were unavailable, published hair identification keys was consulted [22,30–35].

## 2.3 Data analysis

**2.3.1 Abundance and spatiotemporal overlap.** All camera trap images and videos were systematically examined to identify species occurrence. Each detection was classified into an independent event, defined as photographs or videos of the same species taken at the same station with a minimum interval of 30 minutes [36]. Empty frames or records of unidentified organisms were excluded, accounting for <1% of the total dataset. The resulting dataset of independent events formed the basis for estimating species activity patterns, relative abundance, and temporal overlap.

All photograph and video data were processed to assess spatial and temporal overlap among the three-predator species. Spatial overlap among large carnivores was assessed using a Minimum Convex Polygon (MCP) [37] approach implemented in ArcGIS. All camera-trap records were georeferenced, and species-specific detection locations were extracted for spatial analysis. MCPs were constructed to represent the maximum extent of space use for each species that estimate spatial use and overlap, particularly where sample sizes were limited and data were spatially discontinuous [38,39]. Spatial overlap was quantified by calculating intersections among species-specific MCPs, thereby identifying areas of potential co-occurrence within the study landscape. The temporal activity pattern and overlap was quantified using the "Overlap" package in R [40]. Temporal overlap was estimated as the shared area under kernel density curves for two species at any given time point [36]. Following established thresholds, the coefficient of overlap ($\Delta$) was interpreted as: very strong ($\Delta > 0.80$), strong ($0.60 \leq \Delta < 0.80$), moderate ($0.40 \leq \Delta < 0.60$), low ($0.20 \leq \Delta < 0.40$), and very low ($\Delta \leq 0.20$), with slight modification of the framework proposed by Mosquera-Muñoz et al., [41]. Coefficients were estimated using 1,000 bootstrap replicates, with 95% confidence intervals (CIs) derived for robustness. To evaluate circadian activity, all detections were divided into four annual-average ecological time intervals consistent with Central Nepal's photoperiod: dawn (05:00–07:00), day (07:00–17:00), dusk (17:00–19:00), and night (19:00–05:00). This classification follows established protocols [42] and includes partial modifications to align with the average daylight records of the Lapchi Valley.

**2.3.2 Prey consumption analysis.** To estimate prey consumption, scat analyses were employed. The proportion of prey occurrence based solely on hair counts may overestimate the importance of small-bodied species [33,43,44]. Therefore, to correct for this bias, prey biomass consumption was estimated using Ackerman's equation [45]:

$B = 1.98 + 0.035A$

where B is the estimated mass (kg) of prey represented per scat, and *A* is the mean body mass of the prey species. From this, two parameters were calculated:

Biomass consumed (D): $D = B \times C$, where *C* is the number of scats containing the prey species.

Estimated number of prey individuals consumed (E): calculated by scaling biomass consumption against prey body mass.

The proportional contribution of each prey species to leopards' diet was then expressed as:

$$E = \frac{B \times C}{\Sigma(B \times C)} \times 100$$

Average body weights of wild prey species were derived from Shrestha et al., [46], Amin et al., [47], and Karanth and Sunquist [48]. For livestock prey, average body mass values of 80 kg for yak and 125 kg for horse were used, based on reports from local herders regarding depredation losses.

**2.3.3 Dietary overlap.** Additionally, to estimate the dietary overlap between the snow leopards, leopards and wolves, Pianka's index [49], was calculated, which quantifies the degree of similarity in resource use (in this case, prey items) between two species.

$$DO = \frac{\Sigma P_{ij} . P_{ik}}{\sqrt{\left(\Sigma P_{ij}^2\right) . (\Sigma P_{ik}^2)}}$$

Using the formula:

where:

P$_{ij}$ is the proportion of prey category i in the diet of predator j; P$_{ik}$ is the proportion of prey category i in the diet of predator k. The values range between 0 (no overlap) and 1 (complete overlap).

Furthermore, "niche breadth" was measured to estimate the resources used is either generalized or specialized using Standardized Levins' Index (**βα**):

Levins' Index (**β**)

$$\beta = \frac{1}{\Sigma p_i^2}$$

Where,

p$_i$ = proportion of individuals using resource category i.

And

Standardized Levins' Index (**βα**)

**βα** = total number of resource categories. $\beta a = \frac{\beta - 1}{n - 1}$

Where, **βα** ranges from 0 (very specialized) to 1 (very generalized).

## 2.4 Ethical considerations

This study employed exclusively non-invasive methodologies, and no direct handling, capture, or physical interaction with wildlife occurred during the course of the research. All research activities were conducted in strict accordance with the ethical guidelines and legal requirements for wildlife research in Nepal. Prior to data collection, official research approvals were obtained from the Department of National Parks and Wildlife Conservation (DNPWC), Government of Nepal, Kathmandu (Permission Letter No. 35/293/78–79), and the Gaurishankar Conservation Area Project (GCAP) (Permission Letter No. 138/78–79). These approvals ensured that the study complied with national conservation regulations and minimized any potential disturbance to wildlife and their habitats.

## 3. Results

### 3.1 Spatial distribution and overlap

Among the 20 camera trap locations, snow leopards were detected at nine sites within an elevation range of 3,535–4,454 masl, Himalayan wolves were detected at seven sites between 3,535–4,234 masl, and leopards at 11 sites between 2247–4,230 masl (Fig 2). Given the limited number of detections per species, these records represent detection-based occurrences rather than complete species distributions within the landscape. The highest elevation at which all three predators recorded was 4,058 masl. Despite these sampling constraints, the observed elevational patterns suggest broad spatial overlap between snow leopards and Himalayan wolves, with leopards showing greater use of lower and mid-elevation areas. These spatial patterns are consistent with Hypothesis 1, which predicted substantial spatial overlap between snow leopards and Himalayan wolves and partial spatial segregation of leopards along the elevational gradient.

The Minimum Convex Polygon analysis of camera-trap records detections provides an estimate of potential spatial overlap based on observed locations rather than realized or continuous habitat use. Snow leopards were estimated to use an area of 16.88 km², Himalayan wolves 11.33 km², and leopards 59.34 km² within the study area. The MCP representing Himalayan wolf detections was entirely nested within the snow leopard MCP, indicating a strong spatial association in detected use areas. This pattern, however, should be interpreted cautiously given the limited number of wolf detections. In contrast, overlap between snow leopards and leopards was 8.49 km², corresponding to approximately 50.3% of the snow

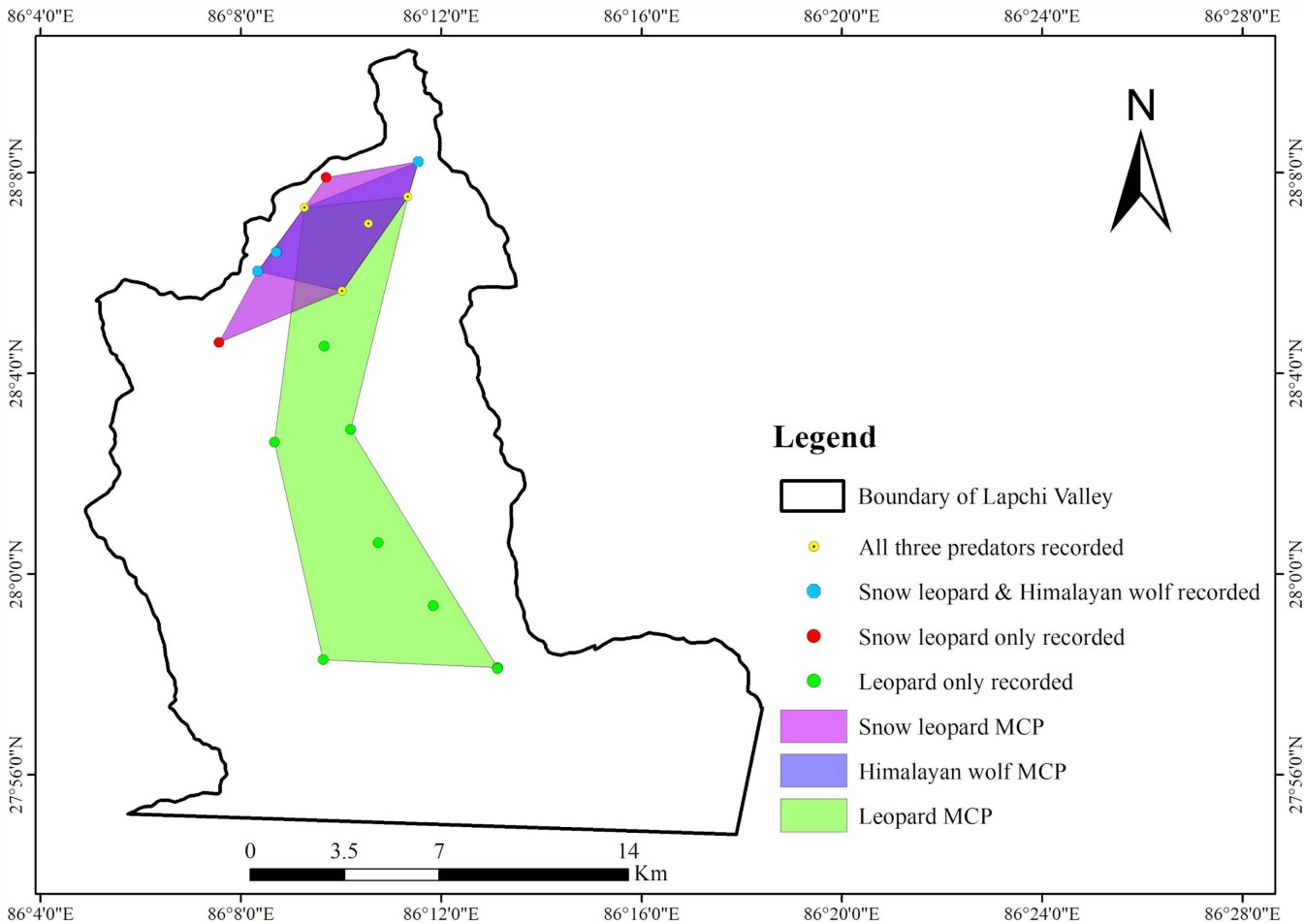

**Fig 2. Minimum Convex Polygon (MCP) analysis showing detection-based spatial extent and overlap of three large carnivores: snow leopards, leopards and Himalayan wolves derived from camera-trap records.**

leopard MCP and 14.3% of the leopard MCP, while overlap between the Himalayan wolf and leopard MCPs was 7.54 km², representing 66.6% of the wolf MCP and 12.7% of the leopard MCP.

### 3.2 Temporal activity pattern and overlap

During the study period a total of 69, 59 and 63 independent activity events were recorded for snow leopards, leopards and Himalayan wolves, respectively. Analysis of temporal activity pattern indicated that all three species exhibited predominantly nocturnal behaviour. The snow leopard demonstrated peak activity at night accounting for 59.42% of recorded events, followed by crepuscular (21.73%) and diurnal (18.84%) periods. Similarly, Himalayan wolves showed strong nocturnal tendencies, with 71.42% of the activities occurring at night, 15.87% during the day and 12.69% during twilight. Leopards also displayed nocturnal dominance (64.40%), followed by diurnal (18.64%) and crepuscular (16.94%) activity (Fig 3).

The temporal overlap analysis revealed a high degree of similarity in activity patterns among the three focal predator species. Across the entire study period, snow leopards exhibited substantial temporal overlap with leopards at 0.85 (95% CI: 0.80–0.96), and Himalayan wolves 0.78 (95% CI: 0.72–0.94). Similarly, leopards and Himalayan wolves showed high overlap with a value of 0.83 (95% CI: 0.77–0.97) (Fig 3).

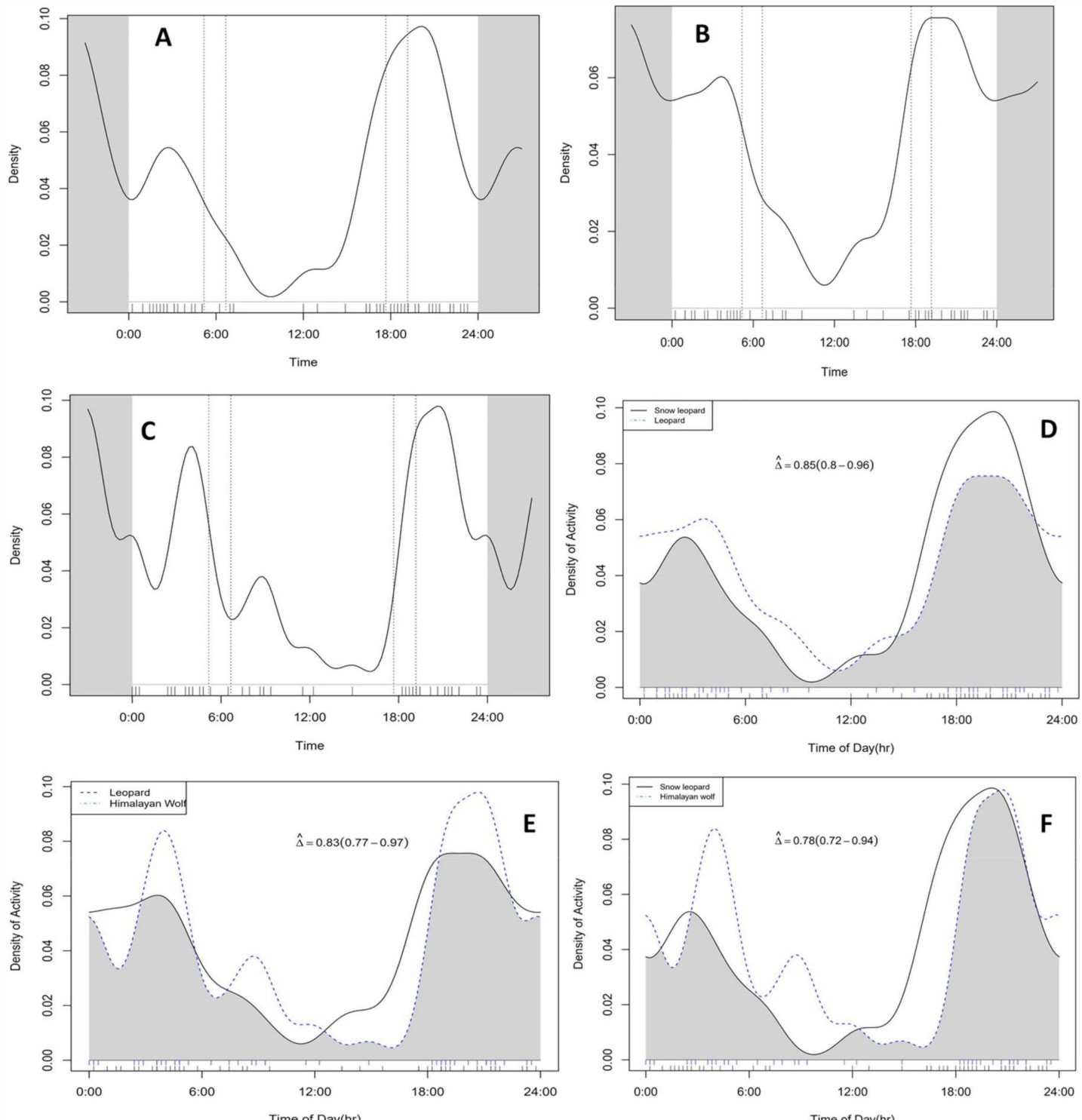

**Fig 3. The temporal activity pattern of snow leopards (A), leopards (B) and Himalayan wolves (C) and the overlapping activity patterns of snow leopards and leopards (D), leopards and Himalayan wolves and snow leopards and Himalayan wolves (F) in Lapchi Valley.**

Seasonal analysis revealed notable variation in temporal overlap among species. During summer, overlap coefficients were generally lower, suggesting increased temporal segregation. Snow leopards showed moderate overlap with leopards (Δ = 0.55, 95% CI: 0.38–0.72) and Himalayan wolves (Δ = 0.59, 95% CI: 0.27–0.90), while leopards and Himalayan wolves exhibited relatively low overlap (Δ = 0.35, 95% CI: 0.13–0.57). In contrast, during winter, higher overlap values were observed across species pairs. Snow leopards showed increased overlap with leopards (Δ = 0.63, 95% CI: 0.52–0.74) and Himalayan wolves (Δ = 0.56, 95% CI: 0.44–0.67), while leopards and Himalayan wolves demonstrated particularly high temporal overlap (Δ = 0.77, 95% CI: 0.65–0.89) (Table 1). These activity patterns are consistent with Hypothesis 2, which predicted predominantly nocturnal behavior and high temporal overlap among the three-predator species.

Furthermore, all three large carnivores exhibited strong temporal overlap with other meso- and large carnivores, including the Himalayan black bear (*Ursus thibetanus*), red fox (*Vulpes vulpes*), and leopard cat (*Prionailurus bengalensis*). In contrast, they showed strong overlap with musk deer (*Moschus leucogaster*), relatively low overlap with key prey species such as yak (*Bos grunniens*), blue sheep (*Pseudois nayaur*), and Assamese macaque (*Macaca assamensis*), while demonstrating moderate temporal overlap with horse (*Equus* sp.), pika (*Ochotona* spp.), Himalayan serow (*Capricornis sumatraensis*), and Himalayan tahr (*Hemitragus jemlahicus*).

### 3.3  Diet composition and overlap

Micro-histological analysis of the fecal samples revealed clear differences in the diets of the three-predator species (Table 2). The snow leopard's diet was dominated by wild prey (totaling 85.15%), with blue sheep contributing 47.14% and musk deer 15.7%. Other wild prey included Himalayan marmot (6.57%), pika, weasel, and unidentified rodents in smaller proportions. Livestock (total 14.55%) remains were also detected, including horse (5.46%), yak (4.58%), and domestic goat (4.51%). In contrast, leopards showed a higher reliance on domestic species (totaling 40.42% of the diet) with wild boar constituting the largest portion of the leopards' diet (35.52%), followed by domestic sheep (13.12%), horse (6.58%), domestic goat (6.15%), and dog (8.82%). Wild prey such as Himalayan tahr, barking deer, and goral were present in lower proportions. The Himalayan wolf exhibited a mixed feeding pattern, with wild prey accounting for a substantial proportion of the diet. Himalayan marmot (32.11%), blue sheep (28.62%), and musk deer (9.57%) were the primary wild prey recorded. Livestock (total of 25.92%) remains were also present, including domestic goat (8.32%), horse (6.14%), and yak (5.81%). These dietary patterns are consistent with Hypothesis 3, which predicted trophic niche differentiation among the three predators, with species-specific reliance on wild prey, livestock, and mixed diets.

### 3.4  Niche breadth

The Standardized Levins' index highlight distinct variation in dietary niche breadth among the three predators. Leopards had the broadest niche (βα = 0.28), followed by Himalayan wolves (βα = 0.22), while snow leopards had the narrowest (βα = 0.17) (Table 3).

Pairwise dietary overlap analyses revealed the highest overlap between snow leopards and Himalayan wolves, Pianka's index = 0.77 [95% CI = 0.718–0.817], indicating substantial similarity in prey consumption between these two species. In contrast, dietary overlap of leopards and snow leopard Pianka's index = 0.087 [95% CI = 0.067–0.108] and between snow leopards and Himalayan wolves (Pianka's index = 0.13 [95% CI = 0.108–0.153]) was comparatively low. These overlap patterns supported the prediction that the strongest dietary overlap between ecologically similar predators, particularly snow leopards and Himalayan wolves.

## 4.  Discussion

Understanding how sympatric apex predators coexist in resource-limited mountain ecosystems is central to both ecological theory and conservation practice. While similar studies have examined pairwise interactions among large carnivores across the Himalayas, the empirical evidence on three co-occurring apex predators within a single, high-elevation pastoral

**Table 1. Temporal overlap of the snow leopards, leopards and Himalayan wolves overall record from camera traps and seasonal overlap.**

| Seasons | Species | Temporal overlapping coefficient (Δ^) [Class interval] | |
|---|---|---|---|
| | | Snow leopard | Leopard |
| Overall | **Leopard** | 0.85 [0.80-0.96] | |
| | **Himalayan wolf** | 0.78 [0.72- 0.94] | 0.83 [0.77-0.97] |
| Summer | **Leopard** | 0.55 [0.38-0.72] | |
| | **Himalayan wolf** | 0.59 [0.27-0.90] | 0.35[0.13-0.57] |
| Winter | **Leopard** | 0.63 [0.52-0.74] | |
| | **Himalayan wolf** | 0.56 [0.44-0.67] | 0.77 [0.65-0.89] |

**Table 2. Diet of snow leopards, leopards and Himalayan wolves: based on proportions (%) of wild and domestic prey species' remains detected in scats.**

| Prey species | Estimated percentage consumption biomass € based on Ackerman's equation | | |
|---|---|---|---|
| | Snow leopard (n=14) | Leopard (n=11) | Himalayan wolf (n=18) |
| Weasel | 2.42 | 0 | 0.36 |
| Unidentified rodent | 2.85 | 1.24 | 1.14 |
| Red fox | 2.14 | 0 | 0 |
| Pika | 3.71 | 0 | 2.17 |
| Musk deer | 15.7 | 1.36 | 9.57 |
| Himalayan tahr | 2.21 | 4.18 | 0 |
| Himalayan marmot | 6.57 | 0 | 28.62 |
| Blue sheep | 47.14 | 1.27 | 32.11 |
| Assamese macaque | 0 | 3.18 | 0 |
| Barking deer | 0 | 7.24 | 0 |
| Wild boar | 0 | 35.52 | 0 |
| Goral | 0 | 5.23 | 0 |
| Unidentified birds | 2.71 | 0.36 | 0.11 |
| Yak # | 4.58 | 1.15 | 5.81 |
| Cattle* | 0 | 4.85 | |
| Domestic goat # | 4.51 | 6.15 | 8.32 |
| Horse # | 5.46 | 6.58 | 6.14 |
| Domestic sheep # | 0 | 13.12 | 5.65 |
| Dog # | 0 | 8.82 | 0 |

# Domestic animals.

**Table 3. The niche breath of the tree predators.**

| Predator | Levins' Index (β) | Standardized Levins' Index (βα) |
|---|---|---|
| Snow leopard | 3.8059 | 0.1651 |
| Leopard | 5.7529 | 0.2796 |
| Himalayan wolf | 4.7159 | 0.2186 |

landscape remains scarce. In this study, we examined spatial, temporal, and trophic interactions among snow leopards, leopards, and Himalayan wolves in the central Himalayan landscape of Nepal. Lapchi Valley represents a distinctive study system characterized by extensive free-ranging livestock grazing, recent leopard expansion into alpine habitats, and strong dependence of local livelihoods on pastoralism [6].

By explicitly testing a priori hypotheses, our findings provide evidence that coexistence among these large carnivores is facilitated primarily through trophic niche differentiation rather than strong spatial or temporal segregation, despite substantial overlap in space and activity. This result highlights diet as the primary axis of niche partitioning in this landscape and illustrates how apex predators may persist under increasing ecological and anthropogenic constraint.

### 4.1 Trophic overlap and differentiation

In the Lapchi Valley, snow leopards primarily consumed wild ungulates, especially blue sheep, whereas leopards depended more heavily on livestock and synanthropic prey such as dogs. Himalayan wolves exhibited a more flexible foraging strategy, exploiting both wild prey and livestock. These contrasting dietary patterns reflect differential coexistence strategies shaped by prey availability, hunting mode, body size, and proximity to human settlements. This trophic segregation is broadly consistent with ecological theory, where sympatric predators partition resources to minimize direct competition [15,50]. This diet partitioning allows the carnivores to reduce direct competition for food by specializing on distinct prey types.

Comparative evidence from other studies highlights the regional variation in predator diets. Previous studies by Lovari et al., (2013a); Weiskopf et al., (2016) and Chetri et al., (2017) found that snow leopards preferentially hunted medium-sized cliff-dwelling ungulates, while wolves targeted plain-dwelling ungulates and small mammals, with limited reliance on livestock. The prevalence of blue sheep in the snow leopards' diet is reflected in the work of Wegge et al., (2012) and Aryal et al., [51]. Leopards' high reliance on domestic prey is consistent with their generalist, opportunistic foraging ecology as observed by Jumabay-Uulu et al., [52] and Lyngdoh et al., [53], likely reflecting reduced wild ungulate densities and accessibility of livestock in valley habitats.

In contrast to several other Himalayan studies, our results indicate a comparatively higher dependence on livestock by Himalayan wolves in Lapchi Valley. This likely reflects local prey scarcity, seasonal livestock availability, and the spatial configuration of grazing areas near wolf-use habitats, particularly along the Nepal–Tibet border. As a subordinate predator relative to snow leopards, wolves may broaden their dietary niche to reduce direct competition, a pattern predicted by intraguild competition theory.

We observed positive correlations in both dietary overlap (Pianka's index = 0.77) and spatial overlap for the snow leopards and Himalayan wolves. Although spatial overlap estimates are detection-based and constrained by sampling effort, the concurrence of high dietary and spatial overlap suggests close ecological connection between these two species. In contract, both dietary and spatial overlap with leopard were lower. Similar pattern have been reported in other carnivora by Azevedo et al., [54]. Lamichhane et al., [55] documented very high dietary overlap between snow leopards and wolves in Shey Phoksundo National Park (Pianka's index = 0.93), with snow leopards consuming more wild prey and wolves heavily dependent on livestock (67.9%), particularly goats and yaks. Leopards appear to hunt at lower elevations, including forested areas near human settlements, paralleling spatial segregation patterns observed in the Indian Trans-Himalayas [56] and Sagarmatha National Park [57,58]. Our findings in Lapchi appear intermediate, suggesting that dietary overlap and intraguild competition is context-dependent, influenced by prey availability, livestock presence, and evolutionary history.

Other studies support this interpretation. Shrotriya et al., (56) showed that wolves, snow leopards, and red foxes in the Trans-Himalaya shared more than 15 food categories, with livestock as a common resource, leading to a mean Pianka's index of 0.503. They concluded that body size and hunting strategies strongly influenced dietary divergence: wolves preferentially selected medium-sized ungulates, snow leopards focused on cliff-dwelling species, and red foxes exploited small prey and anthropogenic resources aligning with our findings. This highlights that carnivore coexistence in

prey-limited mountain ecosystems is mediated by a combination of body size, foraging strategy, and access to anthropogenic subsidies [59].

## 4.2  Habitat preferences cultural context, and competition dynamics

Spatial partitioning further reduces direct competition. Wolves typically forage in valleys and flat terrain near settlements, while snow leopards utilize steep, rugged landscapes suited to ambush hunting [60,61]. In Lapchi, leopards were concentrated at lower elevations, overlapping with snow leopards, wolves and human habitation. This elevational overlap reflects an ongoing range expansion of leopards into alpine systems, likely driven by climate change, forest-line shifts, and human land-use change [6].

Although local perceptions of predators and herder attitudes were not directly assessed in this study, they represent an important socio-cultural dimension shaping coexistence outcomes. Moreover, climate-driven upward shifts of tree-lines, anthropogenic activities, and linear infrastructure development at lower elevation may exacerbate such interactions by bringing leopards into traditional snow leopard range, thus increasing both prey overlap and spatial conflict [6,16]. High spatial and temporal overlap among predators increases the likelihood of livestock depredation, and mis-identification of the responsible predator can exacerbate retaliatory killing—often disproportionately affecting snow leopards, despite leopards being relatively recent arrivals in this system. [6,14,15].

Livestock represents a critical shared resource. In Lapchi, livestock are freely grazed and there is no practice of using containment housing such as corrals and sheds. Here, the diets of both Himalayan wolves and leopards included considerable portions of domestic species (26% and 40% respectively), echoing patterns reported elsewhere [55,56]. This reliance on livestock reflects not only ecological opportunism but also long-standing pastoral land use that has shaped predator–prey systems across the Trans-Himalaya. However increased domestic prey increases depredation risk and fuels retaliatory killings, jeopardizing pastoral livelihoods and predator persistence [12,62–64].

## 4.3  Conservation implications

The coexistence patterns observed in Lapchi Valley offer important conservation insights for multi-predator landscapes experiencing rapid environmental change. First, conserving wild ungulate populations, particularly blue sheep, musk deer, Himalayan serow and Himalayan tahr, is essential to reduce livestock reliance of the predators. Second, conflict mitigation measures including predator-proof corrals, improved husbandry, and transparent and easier compensation schemes are critical to lowering retaliatory killing [57,65].

Our findings indicate that conservation planning must anticipate continued leopard expansion into alpine habitats under climate warming, tree line advance, and changing land use [66,67]. As leopards persist at higher elevations, trophic interactions within the predator guild may change [68]. Leopards, as ecological generalists, are likely to maintain reliance on livestock and synanthropic prey [67], potentially intensifying human–carnivore conflict. In contrast, snow leopards closely tied to rugged alpine terrain and wild ungulates may experience habitat compression if lower elevations become dominated by leopards [4,39]. Himalayan wolves, given their dietary flexibility, may expand into newly available alpine areas or increase reliance on anthropogenic resources where wild prey declines [69]. These differential responses suggest the potential for long-term restructuring of the predator guild. Management strategies should explicitly account for increasing spatial and temporal overlap among apex predators, which may intensify competition and conflict under future climate scenarios [68].

Adaptive land-use planning and protected-area management should integrate elevational range shifts and changing predator interactions [70]. The measures may include safeguarding ecological corridors, maintaining prey-rich refugia, regulating infrastructure expansion along elevation gradients, and applying multi-predator risk assessments in buffer zones. Long-term monitoring of diet, space use, and conflict patterns is essential to detect early signals of guild restructuring and guide proactive, climate-adaptive, species-specific conservation strategies.

By integrating spatial, temporal, and dietary dimensions, this study highlights how niche partitioning enables the persistence of multiple apex predators in a heavily human-influenced alpine system. As climate change accelerates and leopards continue to expand to higher elevations, existing coexistence dynamics are likely to be configured. Management should therefore anticipate these changes and adopt proactive, species-specific strategies in high-elevation ecosystems.

## 5. Conclusion

This study establishes that snow leopards, leopards, and Himalayan wolves coexist in the Lapchi Valley through trophic and ecological partitioning. Snow leopards specialize on wild ungulates, leopards rely on livestock and synanthropic prey, and wolves exhibit dietary plasticity by consuming both wild and domestic species. Even with significant spatial and temporal overlap, these strategies reduce direct competition, enabling multiple apex predators to persist in alpine ecosystems. High spatial and dietary overlap between snow leopards and wolves indicates potential for competitive interactions under prey scarcity or climate-driven habitat contraction, whereas leopards' lower overlap reflects ecological flexibility and ongoing expansion into higher elevations. These dynamics highlight how climate change and anthropogenic pressures may reshape predator guilds and human–wildlife interactions in the Himalaya. Conservation efforts must prioritize for wild prey conservation, particularly blue sheep and musk deer, and conduct community-based measures for livestock management such as predator-proof corrals and effective compensation mechanisms. Overall, the Lapchi Valley illustrates both the resilience and fragility of apex predator coexistence, emphasizing the need for long-term monitoring and adaptive management integrating ecological science with socio-cultural practices.

## Acknowledgments

We would like to thank all the local residents and the Lapchi Monastery for providing meals and accommodation during the fieldwork. We are especially grateful to Karma Sherpa of the Lapchi Monastery Management Committee, Gylazian Sherpa, Shankar Tamang and Gyalung Tamang for their generous support. We also acknowledge the Department of National Parks and Wildlife Conservation (DNPWC), Government of Nepal, Kathmandu, and the Gaurishankar Conservation Area Project (GCAP) for granting research permission as well as the Toledo Zoo for technical assistance. Our sincere thanks go to Peter Zahler, Zoo New England, USA for his valuable assistance in reviewing the manuscript.

## Author contributions

**Conceptualization:** Narayan Prasad Koju, Miriam Lee, Paul Buzzard, Xuelong Jiang, Randall C. Kyes.

**Data curation:** Narayan Prasad Koju, Babita Maharjan, Meghajan Budha, Ashim Thapamagar, Nishan Pokhrel, Paul Buzzard.

**Formal analysis:** Narayan Prasad Koju, Meghajan Budha, Nishan Pokhrel, Laxman Khanal, Paul Buzzard.

**Funding acquisition:** Narayan Prasad Koju, Miriam Lee, Xuelong Jiang.

**Investigation:** Narayan Prasad Koju, Babita Maharjan, Meghajan Budha, Ashim Thapamagar, Miriam Lee, Paul Buzzard, Randall C. Kyes.

**Methodology:** Narayan Prasad Koju, Babita Maharjan, Meghajan Budha, Ashim Thapamagar, Nishan Pokhrel, Paul Buzzard, Randall C. Kyes.

**Project administration:** Narayan Prasad Koju, Miriam Lee, Randall C. Kyes.

**Resources:** Narayan Prasad Koju, Meghajan Budha, Miriam Lee, Randall C. Kyes.

**Software:** Narayan Prasad Koju.

**Supervision:** Narayan Prasad Koju, Xuelong Jiang.

**Validation:** Narayan Prasad Koju, Laxman Khanal, Paul Buzzard.

**Visualization:** Narayan Prasad Koju, Laxman Khanal, Paul Buzzard.

**Writing – original draft:** Narayan Prasad Koju, Miriam Lee, Laxman Khanal, Paul Buzzard, Xuelong Jiang, Randall C. Kyes.

**Writing – review & editing:** Narayan Prasad Koju, Babita Maharjan, Meghajan Budha, Miriam Lee, Laxman Khanal, Paul Buzzard, Xuelong Jiang, Randall C. Kyes.

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
