## [Decision Letter · Decision Letter 0]

16 Dec 2025

Dear Dr. Koju,

Thank you for submitting your manuscript to PLOS ONE. After careful consideration, we feel that it has merit but does not fully meet PLOS ONE’s publication criteria as it currently stands. Therefore, we invite you to submit a revised version of the manuscript that addresses the points raised during the review process.<article class="text-token-text-primary w-full focus:outline-none [--shadow-height:45px] has-data-writing-block:pointer-events-none has-data-writing-block:-mt-(--shadow-height) has-data-writing-block:pt-(--shadow-height) [&:has([data-writing-block])>*]:pointer-events-auto scroll-mt-[calc(var(--header-height)+min(200px,max(70px,20svh)))]" data-scroll-anchor="true" data-testid="conversation-turn-4" data-turn="assistant" data-turn-id="request-WEB:72d4a3d1-613a-42fc-87c4-bb51707507b5-1" dir="auto" tabindex="-1">

The reviewers identified several substantial issues that currently limit the strength of the inferences. In particular, they noted the absence of hypotheses and predictions, insufficient clarity and consistency in the description of the study design and sampling effort, and concerns regarding the robustness of the spatial and temporal analyses. In addition, aspects of the dietary analysis require greater transparency, particularly with respect to sample sizes and uncertainty. The Introduction and Discussion were also considered underdeveloped relative to the scope and importance of the topic, and would benefit from a stronger theoretical framing and clearer articulation of the study’s contribution within the broader literature.

Addressing these points will require significant revisions to multiple sections of the manuscript, including the Introduction, Methods, Results, and Discussion. The reviewers’ detailed comments are provided below and should be carefully considered in preparing a revised version.

plosone@plos.org . A letter that responds to each point raised by the academic editor and reviewer(s). You should upload this letter as a separate file labeled 'Response to Reviewers'.A marked-up copy of your manuscript that highlights changes made to the original version. You should upload this as a separate file labeled 'Revised Manuscript with Track Changes'.An unmarked version of your revised paper without tracked changes. You should upload this as a separate file labeled 'Manuscript'.

We look forward to receiving your revised manuscript.

Kind regards,

Paulo Corti, Ph.D.

Academic Editor

PLOS One

Journal Requirements:

“This study was supported by a faculty grant from the University Grants Commission (UGC), Nepal (FRG 77/78 S&T-01 2020/21 and FRG 080/81 S&T-07), Andrew Sabin Family Foundation, Zoo New England, and the China Exploration and Research Society, Hong Kong. RCK’s effort was supported in part by the Office of Research Infrastructure Programs (ORIP) of the National Institutes of Health through grant number P51OD010425 to the Washington National Primate Research Center, USA”

“This study was supported by a faculty grant from the University Grants Commission (UGC), Nepal (FRG 77/78 S&T-01 2020/21 and FRG 080/81 S&T-07), Andrew Sabin Family Foundation, Zoo New England, and the China Exploration and Research Society, Hong Kong. RCK’s effort was supported in part by the Office of Research Infrastructure Programs (ORIP) of the National Institutes of Health through grant number P51OD010425 to the Washington National Primate Research Center, USA”

4. We note that your Data Availability Statement is currently as follows: [All relevant data are within the manuscript and its Supporting Information files]

5. We note that Figures 1 & 2 in your submission contain [map/satellite] images which may be copyrighted. All PLOS content is published under the Creative Commons Attribution License (CC BY 4.0), which means that the manuscript, images, and Supporting Information files will be freely available online, and any third party is permitted to access, download, copy, distribute, and use these materials in any way, even commercially, with proper attribution. For these reasons, we cannot publish previously copyrighted maps or satellite images created using proprietary data, such as Google software (Google Maps, Street View, and Earth). For more information, see our copyright guidelines: http://journals.plos.org/plosone/s/licenses-and-copyright.

a. You may seek permission from the original copyright holder of Figures 1 &2to publish the content specifically under the CC BY 4.0 license.

USGS EROS (Earth Resources Observatory and Science (EROS) Center) (public domain): http:

Earth (public domain): http://www.naturalearthdata.com/

6. Please remove your figures from within your manuscript file, leaving only the individual TIFF/EPS image files, uploaded separately. These will be automatically included in the reviewers’ PDF.

Reviewers' comments:

Reviewer's Responses to Questions

**Comments to the Author**

1. Is the manuscript technically sound, and do the data support the conclusions?

Reviewer #1: Partly

Reviewer #2: Partly

2. Has the statistical analysis been performed appropriately and rigorously?

Reviewer #1: No

Reviewer #2: N/A

3. Have the authors made all data underlying the findings in their manuscript fully available?

Reviewer #1: Yes

Reviewer #2: Yes

4. Is the manuscript presented in an intelligible fashion and written in standard English?

Reviewer #1: Yes

Reviewer #2: Yes

Reviewer #1: The manuscript addresses an important question about the coexistence of large carnivores. However, the strength of the inferences is limited by (i) the lack of explicit hypotheses/predictions, (ii) inconsistencies and omissions in the design/sampling, and (iii) spatial/temporal conclusions based on very simple methodological decisions without sensitivity analysis. I recommend major revision.

1) Introduction

Hypotheses and predictions are missing. It remains to be clarified what patterns are expected (spatial, temporal, and trophic) and what mechanisms sustain them (competition, exploitation vs. interference, anthropogenic subsidies, prey availability).

I recommend adding hypotheses and predictions

2) Methodology – Design and effort

-Inconsistencies in number of sites/cameras/duration. Twenty-six cameras in 20 locations are mentioned, but Figure 1 shows 19 locations. In addition, the operating time is reported first, followed by the number of cameras (reverse order).

I recommend: (a) standardizing figures in text/figures/tables; (b) describing whether there was >1 camera per location and under what criteria; (c) presenting an effort table with camera days per location and per year, failures, and relocations. The experimental design should be better explained.

-Long study period (2018–2025) without stratification. They report an average of ~761 camera days, which implies that cameras were not active throughout the entire period and possible temporal bias in location.

I recommend: (a) detailing the actual windows of activity for each camera; (b) indicating whether the spatial distribution changed between years; (c) stratifying analyses (or including covariates) by year/season to avoid mixing ecologically dissimilar periods that inflate overlaps.

-Camera model and recording mode. The exact model is requested (line 98). Further on (line 166), it is indicated that there were images and videos, but earlier only images are mentioned.

I recommend specifying the brand/model, sensitivity/intervals, and for each period whether photos, videos, or both were captured, and how they were integrated into the analysis.

3) Methodology – Space and time

-Spatial overlap is calculated using 100 m buffers around detections and altitude limits defined with the same data. This approach is overly simplistic and potentially circular.

I recommend: (a) replacing or supplementing with use estimators (e.g., use density/kernel)); (b) reporting sensitivity analyses to buffer width and altitude limits

-Temporal activity collapsed into 7 years. It is not appropriate to speak of “synchrony” of the three predators with multi-year averaged data.

Request: (a) report Δ with CI and n per pair; (b) stratify by year/season or add a supplementary analysis showing the temporal stability of the patterns.

4) Methodology – Diet

-Sample sizes by species not reported. To interpret trophic overlap, the number of feces analyzed per species (DNA/micro-histology) and the criteria for selecting the histological “subset” are needed.

5) Results

-Temporal synchrony (l. 260–261): Avoid statements of synchrony for the three predators without temporal stratification.

-Spatial overlap: Strong conclusions are based on buffer + altitudinal clipping.

-Diet: Include n per species analyzed and Pianka's CI (bootstrap per scat).

To support the Discussion and conclusions, the manuscript should: (i) state hypotheses/predictions and mechanisms; (ii) clarify and tabulate design/effort (with temporal stratification); (iii) replace or, at least, subject to sensitivity analysis the spatial analysis based on buffers; and (iv) strengthen the diet section with n per species. These improvements are necessary to ensure reproducibility and inferential validity.

Reviewer #2: Comments to authors

I read with great interest your article titled “Niche Partitioning Facilitates Coexistence of Three Apex Predators in the Lapchi Valley, Central Himalaya, Nepal.” This study examines niche partitioning and coexistence among sympatric snow leopards, leopards, and Himalayan wolves in the Lapchi Valley of Central Nepal. Using camera trapping and dietary analyses, the authors assess spatial, temporal, and trophic overlap to explore mechanisms facilitating predator coexistence in a high-altitude Himalayan system. My primary concerns are related to some of the methods and the depth of the introduction and discussion given the nature of the topic and approaches.

Please see my specific line comments below.

Abstract

In the abstract, it would be helpful to start with one or two sentences that provide some context and clearly point out the main gap before moving into the study objectives.

Introduction

The introduction currently reads thin and would benefit from further development. Lines 47–54 discuss the threats these apex predators face, but the content is split across two paragraphs and would benefit from better organization.

While some of these points were briefly mentioned, the authors should further expand on contexts such as the role of apex predators in the ecosystem, existing mechanisms of coexistence among apex predators (e.g., different types of niche partitioning), the remaining knowledge gaps, and what makes the study system unique, etc. Expanding on these areas would provide a stronger theoretical background for the study.

Methods

Please ensure consistent citation formatting.

The spatial overlap analysis adopts a presence-based, rule-constrained approach, conceptually similar to an environmental envelope method, rather than more complex approaches such as MCP, SDM, or JSDM. The authors should more explicitly justify their methodological choice and discuss its underlying assumptions, rather than adopting alternative approaches.

Please ensure that index abbreviations are used consistently across the manuscript (e.g., “B” is used twice for different indices).

Results

The spatial overlap results are derived from a relatively small number of camera trap sites (7–11 detections per species). Given this limited spatial sampling, stronger justification is needed when interpreting these estimates, particularly statements suggesting complete spatial overlap or subsumed distributions. The authors should clarify that these results reflect detection-based, potential overlap, and discuss how sampling design and buffer choices may influence the observed patterns, or alternatively consider using additional data or complementary methods to support their conclusions.

Please ensure consistent table formatting. Please also correct the grammar in lines 299–301; there is an extra closing parenthesis.

Discussion

Please ensure consistent citation formatting throughout the manuscript.

The Discussion should be expanded and developed in greater depth, particularly with respect to the conservation implications of the findings. Given that both the topic and the analytical approaches are not particularly novel, and that similar analyses of apex and meso-predator interactions have been conducted across other regions of the Himalayas, it is especially important for the authors to clearly articulate how this study system differs from previous work and what new insights it provides. In particular, the authors should better explain why understanding niche partitioning among snow leopards, leopards, and Himalayan wolves in this specific area matters. For example, do the observed dietary patterns reflect differential coexistence strategies, human-predator relationships, or local perceptions of predators by herders? What about the spatial overlap results, and how do those ecological findings link to the cultural context? What are the potential long-term implications for prey resources, habitat use, or conflict risk given the high spatial and temporal overlap observed among these species?

**Do you want your identity to be public for this peer review?** For information about this choice, including consent withdrawal, please see our Privacy Policy

Reviewer #1: No

Reviewer #2: No

---

## [Author Response · Author response to Decision Letter 1]

16 Jan 2026

PONE-D-25-54236

Comments from Editor

Response: Thank you for support and consideration. We have carefully reviewed and revised the manuscript to ensure full compliance with PLOS ONE’s formatting and style requirements. The official PLOS ONE templates for the main text and for the title, authors, and affiliations were used. File naming conventions have also been corrected accordingly.

Response: Thank you for the guidance. We have revised the manuscript to fully comply with PLOS ONE’s formatting and style requirements, including use of the official templates and correct file naming.2. Thank you for stating in your Funding Statement:

“This study was supported by a faculty grant from the University Grants Commission (UGC), Nepal (FRG 77/78 S&T-01 2020/21 and FRG 080/81 S&T-07), Andrew Sabin Family Foundation, Zoo New England, and the China Exploration and Research Society, Hong Kong. RCK’s effort was supported in part by the Office of Research Infrastructure Programs (ORIP) of the National Institutes of Health through grant number P51OD010425 to the Washington National Primate Research Center, USA”

Response: We thank the Editor for this comment. The Funding Statement has been amended to clearly declare all sources of financial support received during the study following the feedback.

“This study was supported by a faculty grant from the University Grants Commission (UGC), Nepal (FRG 77/78 S&T-01 2020/21 and FRG 080/81 S&T-07), Andrew Sabin Family Foundation, Zoo New England, and the China Exploration and Research Society, Hong Kong. RCK’s effort was supported in part by the Office of Research Infrastructure Programs (ORIP) of the National Institutes of Health through grant number P51OD010425 to the Washington National Primate Research Center, USA”

Response: The Funding Statement has been amended and statement "The funders had no role in study design, data collection and analysis, decision to publish, or preparation of the manuscript" is added.

4. We note that your Data Availability Statement is currently as follows: [All relevant data are within the manuscript and its Supporting Information files]

Response: We confirm that our submission contains the minimal data set required to replicate all results reported in the study. All supplementary filed, associated metadata and R code have been deposited in Zenodo. The data and code are publicly available at: https://doi.org/10.5281/zenodo.18053588 . The Data Availability Statement has been updated accordingly.

5. We note that Figures 1 & 2 in your submission contain [map/satellite] images which may be copyrighted. All PLOS content is published under the Creative Commons Attribution License (CC BY 4.0), which means that the manuscript, images, and Supporting Information files will be freely available online, and any third party is permitted to access, download, copy, distribute, and use these materials in any way, even commercially, with proper attribution. For these reasons, we cannot publish previously copyrighted maps or satellite images created using proprietary data, such as Google software (Google Maps, Street View, and Earth). For more information, see our copyright guidelines: http://journals.plos.org/plosone/s/licenses-and-copyright.

a. You may seek permission from the original copyright holder of Figures 1 &2to publish the content specifically under the CC BY 4.0 license.

USGS EROS (Earth Resources Observatory and Science (EROS) Center) (public domain): http:

Earth (public domain): http://www.naturalearthdata.com/

Response: Thank you for raising this copyright concern. The Figure 1 was generated by the authors using the ESRI Global Land Cover dataset obtained from the ESRI Living Atlas (https://livingatlas.arcgis.com/landcover/). This dataset is distributed under the Creative Commons Attribution (CC BY 4.0) license, and no proprietary sources (e.g., Google Maps/Earth) were used. The figure caption has been updated to include proper attribution.

Figure 2 was created using a Digital Elevation Model (DEM) downloaded from USGS (http://viewer.nationalmap.gov/viewer/), which is publicly available and compatible with the CC BY 4.0 license. In addition, Figure 2 has been revised in the revised manuscript following Reviewer 2’s comments. Accordingly, both figures comply with PLOS ONE’s CC BY 4.0 licensing requirements, therefore no additional permissions are required.

6. Please remove your figures from within your manuscript file, leaving only the individual TIFF/EPS image files, uploaded separately. These will be automatically included in the reviewers’ PDF.

Response: All figures are submitted in TIFF and removed from manuscript file as per comment in revised submission.

Response: There is no specific citation recommended by the reviewers.

Reviewers' comments:

Reviewer #1:

General comment: The manuscript addresses an important question about the coexistence of large carnivores. However, the strength of the inferences is limited by (i) the lack of explicit hypotheses/predictions, (ii) inconsistencies and omissions in the design/sampling, and (iii) spatial/temporal conclusions based on very simple methodological decisions without sensitivity analysis. I recommend major revision.

1) Introduction

Hypotheses and predictions are missing. It remains to be clarified what patterns are expected (spatial, temporal, and trophic) and what mechanisms sustain them (competition, exploitation vs. interference, anthropogenic subsidies, prey availability).

I recommend adding hypotheses and predictions

Response: We thank the reviewer for this constructive and insightful comment. We agree that explicitly stating hypotheses and predictions is essential to strengthen inference and clarify the analytical framework. In response, we have substantially revised the Introduction to frame the study as hypothesis-driven and to explicitly link our analyses to ecological mechanisms underlying carnivore coexistence. Precisely, we added a new paragraph at the end of the Introduction that clearly pronounces hypotheses and predictions related to spatial, temporal, and trophic niche partitioning among snow leopards, leopards, and Himalayan wolves.

2) Methodology – Design and effort

-Inconsistencies in number of sites/cameras/duration. Twenty-six cameras in 20 locations are mentioned, but Figure 1 shows 19 locations. In addition, the operating time is reported first, followed by the number of cameras (reverse order).

I recommend: (a) standardizing figures in text/figures/tables; (b) describing whether there was >1 camera per location and under what criteria; (c) presenting an effort table with camera days per location and per year, failures, and relocations. The experimental design should be better explained.

-Long study period (2018–2025) without stratification. They report an average of ~761 camera days, which implies that cameras were not active throughout the entire period and possible temporal bias in location.

I recommend: (a) detailing the actual windows of activity for each camera; (b) indicating whether the spatial distribution changed between years; (c) stratifying analyses (or including covariates) by year/season to avoid mixing ecologically dissimilar periods that inflate overlaps.

-Camera model and recording mode. The exact model is requested (line 98). Further on (line 166), it is indicated that there were images and videos, but earlier only images are mentioned.

I recommend specifying the brand/model, sensitivity/intervals, and for each period whether photos, videos, or both were captured, and how they were integrated into the analysis.

Response: We thank the reviewer for highlighting these important points. In response:

1. Camera trap locations: The missing camera trap has been added in the revised Figure 1, ensuring consistency between the text and figure.

2. Camera trap effort and design: We have added a supplementary table detailing camera trap installation dates, active periods, removal dates, and total working hours for each camera. This table provides camera days per location and per year, including information on camera failures and relocations, clarifying the experimental design and effort across the study period.

3. Camera model and settings: We used Bushnell Trophy Cam HD Trail Cameras (Essential E3 16MP and E2 12MP). Cameras were set to hybrid mode (three photos followed by a 10-second video) with a trigger interval of 1 second and sensitivity set to low. These settings were applied consistently across the study period, and both images and videos were integrated into the analysis.

3) Methodology – Space and time

-Spatial overlap is calculated using 100 m buffers around detections and altitude limits defined with the same data. This approach is overly simplistic and potentially circular.

I recommend: (a) replacing or supplementing with use estimators (e.g., use density/kernel)); (b) reporting sensitivity analyses to buffer width and altitude limits

Response: We are thankful for both reviewers for the suggestion. Following the reviewers’ (including reviewer #2) suggestions, we applied Minimum Convex Polygon (MCP) analysis to more robustly quantify the spatial distribution and overlap of the three predators. During MCP construction, we accounted for elevation and land cover, including high steep mountain areas, to ensure realistic habitat representation. This approach supplements our previous presence-based method, reducing potential circularity from using the same data to define buffers and altitude limits. We have updated the Results and Discussion sections to reflect these revised analyses.

-Temporal activity collapsed into 7 years. It is not appropriate to speak of “synchrony” of the three predators with multi-year averaged data.

Request: (a) report Δ with CI and n per pair; (b) stratify by year/season or add a supplementary analysis showing the temporal stability of the patterns.

Response: We thank the reviewer for highlighting this important point. In response, we have reported Δ values with confidence intervals (CI) and sample sizes (n) for each predator pair, which are now presented in Table 1 and Table 2. Additionally, to address concerns regarding temporal averaging, we have stratified the analyses by season and included a supplementary analysis showing the seasonal

---

## [Decision Letter · Decision Letter 1]

15 Feb 2026

Dear Dr. Koju,

https://journals.plos.org/plosone/s/submission-guidelines#loc-laboratory-protocols . Additionally, PLOS ONE offers an option for publishing peer-reviewed Lab Protocol articles, which describe protocols hosted on protocols.io. Read more information on sharing protocols at https://plos.org/protocols?utm_medium=editorial-email&utm_source=authorletters&utm_campaign=protocols .

We look forward to receiving your revised manuscript.

Kind regards,

Paulo Corti, Ph.D.

Academic Editor

PLOS One

Journal Requirements:

Reviewers' comments:

Reviewer's Responses to Questions

**Comments to the Author**

Reviewer #1: All comments have been addressed

Reviewer #2: All comments have been addressed

2. Is the manuscript technically sound, and do the data support the conclusions?

Reviewer #1: Yes

Reviewer #2: Yes

3. Has the statistical analysis been performed appropriately and rigorously?

Reviewer #1: Yes

Reviewer #2: Yes

4. Have the authors made all data underlying the findings in their manuscript fully available?

Reviewer #1: (No Response)

Reviewer #2: Yes

5. Is the manuscript presented in an intelligible fashion and written in standard English?

Reviewer #1: Yes

Reviewer #2: Yes

Reviewer #1: (No Response)

Reviewer #2: I read with great interest your article titled “Niche Partitioning Facilitates Coexistence of Three Apex Predators in the Lapchi Valley, Central Himalaya, Nepal.” The authors have effectively addressed the reviewers’ comments from the previous review round. I only have a few minor comments remaining.

Lines 49-50: Classification should be specified as by the IUCN Red List.

Lines 54-61: These sentences provide important background information about the study species, but they feel out of place after the knowledge gap has already been introduced. I suggest moving them earlier in the paragraph right after introducing snow leopards and Himalayan wolves as the dominant alpine carnivores. The paragraph could then follow a clearer logical flow: (1) Snow leopards and Himalayan wolves are the dominant carnivores in this alpine system, followed by one or two sentences describing each species; (2) in recent years, leopards have increasingly expanded into snow leopard habitat, followed by one or two sentences introducing leopards; (3) this recent range expansion (currently line 47) then naturally leads into the knowledge gap.

Line 125 and multiple instances in Discussion: Please ensure consistency in citation formatting, particularly in the use of parentheses versus brackets.

Line 277: The abbreviation of MCP has already been noted earlier in line 204。

Discussion: The conservation implications section could be strengthened by being more specific, as the current recommendations feel somewhat generic. It would also benefit from more explicitly engaging with the broader consequences of ongoing leopard range expansion and climate-driven habitat shifts. For example, do the authors expect to see similar or different range or diet shifts in snow leopards and wolves as well? What might this imply for longer-term restructuring of the predator guild under continued environmental change? How might future land-use planning or protected-area management adapt to these elevational range shifts and changing predator interactions? Expanding this section to address these forward-looking and differential management considerations would enhance the applied relevance of the study, especially in the context of rapid climate change in high-elevation ecosystems.

**Do you want your identity to be public for this peer review?** For information about this choice, including consent withdrawal, please see our Privacy Policy

Reviewer #1: No

Reviewer #2: No

---

## [Author Response · Author response to Decision Letter 2]

17 Feb 2026

Response to Reviewers’ Comments to the Author

Comment from editor: 1. Please ensure that you include a legends for Figure 1 to 3 within your main document. We do appreciate that you have a figure legends document uploaded as a separate file, however, we do require this to be part of the manuscript file itself and not uploaded separately.

Reply: All figures legends are added in main document.

Reviewer #1: (No Response)

Reply: Thank you for support and encouragement

Reviewer #2: I read with great interest your article titled “Niche Partitioning Facilitates Coexistence of Three Apex Predators in the Lapchi Valley, Central Himalaya, Nepal.” The authors have effectively addressed the reviewers’ comments from the previous review round. I only have a few minor comments remaining.

Reply: We greatly appreciate your continued support and constructive feedback. Your insightful comments have been invaluable in helping us further refine the manuscript and enhance its scientific accuracy. We are grateful for your suggestions, which have contributed significantly to improving the clarity, robustness, and overall quality of our manuscript.

Comment 1: Lines 49-50: Classification should be specified as by the IUCN Red List.

Reply: The classification of the species has been revised to explicitly indicate that it follows the IUCN Red List, as recommended

Lines 54-61: These sentences provide important background information about the study species, but they feel out of place after the knowledge gap has already been introduced. I suggest moving them earlier in the paragraph right after introducing snow leopards and Himalayan wolves as the dominant alpine carnivores. The paragraph could then follow a clearer logical flow: (1) Snow leopards and Himalayan wolves are the dominant carnivores in this alpine system, followed by one or two sentences describing each species; (2) in recent years, leopards have increasingly expanded into snow leopard habitat, followed by one or two sentences introducing leopards; (3) this recent range expansion (currently line 47) then naturally leads into the knowledge gap.

Reply: We thank the reviewer for this constructive suggestion. In response, we have revised the first and second paragraphs of the Introduction to improve the logical flow. The background information on the study species has been repositioned immediately after introducing snow leopards and Himalayan wolves, followed by details on leopards and their recent range expansion.

Line 125 and multiple instances in Discussion: Please ensure consistency in citation formatting, particularly in the use of parentheses versus brackets.

Reply: All citations in the manuscript, including line 125 and multiple instances in the Discussion, have been carefully revised to ensure consistency and compliance with the journal’s formatting guidelines.

Line 277: The abbreviation of MCP has already been noted earlier in line 204。

Reply: Deleted the abbreviation

Discussion: The conservation implications section could be strengthened by being more specific, as the current recommendations feel somewhat generic. It would also benefit from more explicitly engaging with the broader consequences of ongoing leopard range expansion and climate-driven habitat shifts. For example, do the authors expect to see similar or different range or diet shifts in snow leopards and wolves as well? What might this imply for longer-term restructuring of the predator guild under continued environmental change? How might future land-use planning or protected-area management adapt to these elevational range shifts and changing predator interactions? Expanding this section to address these forward-looking and differential management considerations would enhance the applied relevance of the study, especially in the context of rapid climate change in high-elevation ecosystems.

Reply: We thank the reviewer for this valuable suggestion. In response, Section 4.3 on ‘Conservation Implications’ has been considerably revised and strengthened. We have incorporated more specific recommendations, clearly addressing the potential consequences of ongoing leopard range expansion, climate-driven habitat shifts, and possible responses of snow leopards and Himalayan wolves. Additionally, we discuss the implications for long-term restructuring of the predator guild and provide considerations for adaptive land-use planning and protected-area management under changing high-elevation ecosystems.

---

## [Editor Report · Decision Letter 2]

26 Feb 2026

Niche Partitioning Facilitates Coexistence of Three Apex Predators in the Lapchi Valley, Central Himalaya, Nepal

PONE-D-25-54236R2

Dear Dr. Koju,

We’re pleased to inform you that your manuscript has been judged scientifically suitable for publication and will be formally accepted for publication once it meets all outstanding technical requirements.

Kind regards,

Paulo Corti, Ph.D.

Academic Editor

PLOS One

---

## [Editor Report · Acceptance letter]

PONE-D-25-54236R2

PLOS One

Dear Dr. Koju,

I'm pleased to inform you that your manuscript has been deemed suitable for publication in PLOS One. Congratulations! Your manuscript is now being handed over to our production team.

Kind regards,

on behalf of

Dr. Paulo Corti

Academic Editor

PLOS One